# In Vivo (In)Stability Shoulder Assessment in Healthy Active Adults Using Force Plates and a Motion Capture System: A Cross-Sectional Study

**DOI:** 10.3390/s25175333

**Published:** 2025-08-27

**Authors:** Laura Ramírez-Pérez, Eric Yung-Sheng Su, Antonio Ignacio Cuesta-Vargas, Graham K. Kerr

**Affiliations:** 1Department of Physiotherapy, Clinimetric Group F-14, The Institute of Biomedical Research in Málaga (IBIMA, BIONAND Platform), University of Málaga, C/Arquitecto Francisco Peñalosa 3, 29071 Málaga, Spain; acuesta@uma.es; 2School of Exercise and Nutrition Sciences, Faculty of Health, ARC ITTC for Joint Biomechanics, Centre for Biomedical Technology, Queensland University of Technology, 2 George St, Brisbane City, QLD 4000, Australia; eric.su@qut.edu.au (E.Y.-S.S.); g.kerr@qut.edu.au (G.K.K.)

**Keywords:** shoulder, stability, force plates, motion capture system, assessment, strength

## Abstract

The assessment of shoulder stability is a great challenge in sports medicine. There is a lack of objective tools to assess functional shoulder stability in sports with high demands on the upper limb. This cross-sectional study recruited twenty healthy adults to analyze the use of a force platform in a push-up analysis as a valid tool for estimating glenohumeral stability. For this purpose, the subjects performed one strength task based on a maximum lateral abduction against a dynamometer. They also performed three variations of a push-up task on force plates with movements recorded by a 3D motion capture system. The results showed that healthy adults present similar movement patterns during push-ups, without differences in terms of stability between sexes, although males showed greater values in lateral abduction strength (left: 63.2 vs. 36.8; *p* < 0.001; right: 64.2 vs. 38.9; *p* < 0.001) and ground reaction force peak in the three push-up tasks (*p* < 0.005). Moreover, four prediction models were developed based on the use of force plate data to estimate kinematics concerning humerus acceleration (*p* < 0.001). In conclusion, this research demonstrated that force plates are a valid tool for upper-limb assessment with significant correlations with dynamometer and 3D motion capture measures.

## 1. Introduction

The shoulder is the most complex joint in the body because it involves five different joints, with a large number of ligaments and muscles involved in the union of the axial skeleton with the upper extremities, and it is critically important for the use of arms and hands in daily life [1]. Nonetheless, the complex structure and the great freedom of movement predispose the glenohumeral joint to instability [2]. Glenohumeral instability is a very common pathology [3], with a higher incidence in high-performance athletes. It is also a major concern in sports that involve throwing or repetitive motions where the elbow is above the shoulder height since the biomechanics of these motions require high demands on static and dynamic stabilizers of the glenohumeral complex [4]. Moreover, this pathology is the most common cause of time off from sports, and not all injured athletes achieve complete readaptation to the same competition level prior to injury [5].

Traditionally, the problem of not achieving a complete recovery has been attributed to structural damage [6]. However, it has been demonstrated that abnormal muscle activation patterns of the rotator cuff, shoulder periarticular muscles, and the large thoracic muscles play a relevant role in this injury [7]. In addition, peripheral and central fatigue, as well as psychological variables, together with the absence of good neuromuscular coordination, may affect the capacity to return to play in high-performance athletes [8,9,10].

Regarding the prior evaluation to estimate glenohumeral instability, physical tests such as the “relocation test”, “Drawer test”, “sulcus test”, or “apprehension test” have been used [11]. In addition, objective measurements have been used due to their high relationship with joint stability, such as strength, evaluated with handheld and isokinetic dynamometers [12,13,14]; scapulohumeral kinematics, assessed with high-speed cameras [15]; velocity loss during repetitive exercise, analyzed with linear velocity transducers [16]; and motor control and physiological fatigue, evaluated with electromyography [17,18]. Nonetheless, there is no consistent evidence concerning objective tests to determine the type of exercises that athletes can endure without suffering an episode of instability [18].

Currently, the use of force platforms is becoming more widespread in physiotherapy evaluations, and several authors have started to use them to assess sensorimotor control [19,20]. Furthermore, in regard to the evaluation tasks, push-ups are one of the most complex movements in terms of shoulder stability [21] due to the diversity of muscle activation patterns that this movement presents, in which loads, force, and power can also be quantified [22].

For these reasons, the main objective of the present study is to validate the use of a force platform in a push-up analysis as a valid tool for estimating glenohumeral stability.

## 2. Materials and Methods

### 2.1. Study Setting and Participants

This cross-sectional study recruited twenty participants from the University of Queensland and Queensland University of Technology (Brisbane, Australia). The participants were English speakers over 18 years of age who carried out physical activity regularly and signed the informed consent. Participants were excluded if they had a history of shoulder instability or reported pain suspected to originate from any intra-articular, tendon, or shoulder muscle injury, pain following an acute trauma, or other current musculoskeletal injury that had limited shoulder function or caused them to seek care within the previous six months. Furthermore, participants were excluded if they had any history of rheumatoid, inflammatory, or neurological disease, surgery to the back or upper limbs, or if they were physically unable to perform the testing procedures.

This study received ethics approval from the University Human Research Ethics Committee of the Queensland University of Technology (approval number: 8354). Moreover, the principles of the Declaration of Helsinki were respected, and the research team guaranteed quality management as per the definition in DIN ISO 9000 (DIN, 2015) [23].

### 2.2. Equipment and Outcome Measures

#### 2.2.1. Clinical Assessment

An experienced physiotherapist developed three clinical tests to determine if the participants had any shoulder stability problems. The tests performed were anterior and posterior Drawer tests, in which the physiotherapist stabilizes the acromion with one hand and performs an anterior or posterior movement of the humeral head in the glenoid cavity, and load and shift tests, in which the physiotherapist stabilizes the scapula with one hand and applies a load and shift of the humeral head in an anteromedial direction [11].

#### 2.2.2. Kinetics (Force)

An isokinetic dynamometer, Biodex System 3 (Biodex Medical Systems, New York, NY, USA), was used to record strength data in isometric mode. The dynamometer has shown excellent reliability (Intraclass Correlation Coefficient → ICC > 0.75; Standard Error of Measurement → SEM = 3.8 Nm) in measuring shoulder force [24]. The participants were seated with their arms abducted at 90 degrees and performed a maximum isometric lateral abduction for 3 repetitions with 30 s of rest between attempts. This was performed on the right and left arms separately. The main outcome variables extracted were peak force, mean force across the three trials, and time to peak force.

The system was calibrated prior to testing, following the manufacturer’s standard procedure, and torque data were sampled at 1000 Hz.

#### 2.2.3. Push-Up Tasks

The participants performed three different push-up tests using in-ground force plates model 0R6-7, AMTI (Advanced Mechanical Technology Inc., Waterton, MA, USA). The force plates have been reported to have excellent reliability in measuring counter-movement push-up forces (ICC = 0.84–0.96; SEM = 31.1 N) [25]. This instrument was used during the following three tests:Static push-up test: Maintain the push-up position with the hands shoulder-width apart, elbows completely extended, feet touching the floor with the knees extended, and the hip in neutral position for thirty seconds.Countermovement push-up test: Perform the maximum possible number of push-ups in thirty seconds.Jumping push-up test: Carry out three repetitions of the jumping push-up.

To standardize the push-up technique, the subjects were instructed with the following orders:-Maintain a straight body posture, forming a line across the head, shoulders, hips, knees, and feet.-Put your hands shoulder-width apart, with the fingers facing forward.-Place the feet hip-width apart, with the toes touching the floor, forming a right angle at the ankles.-Perform the push-ups, flexing the elbow until 90º degrees, and return to the starting position.-Avoid movement of the back, pelvis, and knees during the tasks.-Remove both hands from the force plates at the same time during the jumping push-ups.

The system was calibrated before each data collection, together with the motion capture system, to ensure the synchronization process, and the data were sampled at 1000 Hz.

Outcome variables were the number of push-ups performed, displacement of the center of pressure in both the anteroposterior and mediolateral axes, and magnitude of peak ground reaction force.

#### 2.2.4. Kinematics

A 12-camera Vicon Vero v2.2. three-dimensional motion capture system (Vicon MX, Oxford, UK) was used to track the motions of body segments for movement analysis due to their excellent reliability (ICC = 0.9; SEM = 1.9 degrees) in terms of dynamic upper-limb motion, including push-ups [26]. A total of 44 passive reflective markers were attached to the participant on key body landmarks, defining the joint coordinate system based on the International Society of Biomechanics Recommendations [27]. Based on previous studies [28], three additional clusters of four markers were attached to the acromion and interscapular region to track kinematics more accurately (Figure 1). The 56 markers are located as follows:Spine/thorax: The 7th cervical vertebrae, 10th thoracic vertebrae, jugular space, xiphoid process, and one cluster on the interscapular region.Upper limb: The 2nd knuckle, 3rd knuckle, 5th knuckle, radial styloid, ulnar styloid, ½ surface of the forearm, lateral epicondyle, medial epicondyle, ½ surface of the upper arm, and one cluster on the acromioclavicular joint.Pelvis: The ½ iliac crest, posterosuperior iliac spine, and greater trochanter.Lower limb: The 5th metatarsophalangeal joint, posterior surface of the calcaneus, tibial malleolus, peroneal malleolus, ½ surface of the shank, lateral side of the tibial plateau, medial side of the tibial plateau, and ½ surface of the thigh.The marker positions used in this study were selected following the standards of the International Society of Biomechanics [27], in conjunction with the protocols provided by the manufacturer of the three-dimensional motion capture system. Therefore, the markers are not placed directly on muscles but on bony landmarks to minimize soft tissue motion artifacts that could lead to errors in the joint kinematics analysis. Moreover, these points should be anatomically relevant to define shoulder or thorax segments, and should not be occluded during the selected task to avoid the risk of tracking errors.

Cameras recorded the dynamic tasks executed on force plates, extracting outcome variables of joint angles, acceleration, deceleration, and percentage of decrement of acceleration of the markers located on the humerus and the forearm during the push-up tasks.

The system was dynamically calibrated prior to each session, ensuring a reconstruction error below 0.5 mm using a sampling frequency of 200 Hz, synchronizing the data with force plate data using Vicon Nexus (version 2.15.0. x64).

### 2.3. Data Analysis

All dynamic time-varying data were collected using the software provided by the manufacturing companies. Before signal processing, data were filtered using a second-order Butterworth low-pass filter with a cut-off frequency of 8 Hz for torque signals, 15 Hz for ground reaction force and moment signals, and 8 Hz for marker trajectories. Then, data were processed using a combination of the company software and custom-written code in MATLAB version R2024a (Mathsworks Inc., Englewood Cliffs, NJ, USA).

Regarding statistics, descriptive data (mean, standard deviation, and confidence interval) were calculated for all variables using standard procedures. Furthermore, the effect size was calculated between the sexes.

To represent the data graphically, median values and envelopes were calculated for the entire sample. To allow for direct comparison across participants despite differences in execution speed, both the counter-movement push-up (Figure 2) and the jumping push-up (Figure 3) were time-normalized. For each trial, movement onset and termination were identified using the vertical displacement of the humerus marker. The data were then interpolated to 101 equally spaced points, representing 0–100% of the task duration. This temporal normalization allowed the computation of group median curves and variability envelopes to reflect the movement pattern while preserving amplitude differences.

The Shapiro–Wilk test was used to assess normality of the data, obtaining no significant deviation from normality (*p* > 0.05). Therefore, Pearson’s correlation analysis was performed between the values obtained by force plates and a three-dimensional motion capture system. Moreover, dynamometric measurements were correlated with both force plates and camera data to correlate kinetic and kinematic measurements.

An ANOVA estimation model was calculated using weighted multivariate linear regression analysis to try to establish prediction equations, based on the use of the force plates data to estimate kinematics variables, specifically the acceleration of the humerus.

For all statistical comparisons, the α level was set at 0.05. Jamovi 2.4.11. The jamovi project (Jamovi Stats, Sydney, Australia) was used for all statistical computations.

### 2.4. Sample Size

To carry out the study, the sample size was calculated using an a priori analysis establishing a large correlation of 0.60, a statistical power of 0.90, and an alpha error of 0.05, resulting in a minimum sample size of seventeen healthy active adults.

The sample size was calculated using the software GPower 3.1.9.7 (University of Düsseldorf, Germany).

## 3. Results

The total sample comprised 20 healthy adults, with 14 males (70%) and 6 females (30%), with an average age of 32.7 ± 7.6 years and a body mass index of 23.8 ± 3.17. The participants were from 10 countries: Spain (5), France (4), Australia (3), Italy (2), New Zealand (1), the Netherlands (1), Morocco (1), Sri Lanka (1), India (1), and Taiwan (1).

In addition, in terms of stability, all patients were negative in the anterior and posterior Drawer tests, as well as in the load and shift test, meaning they did not have any shoulder instability.

Table 1 summarizes the descriptive analysis of the strength data obtained from the dynamometer in the maximum isometric lateral abduction test performed in both upper limbs. Males had higher peak lateral abduction force than females in both the left shoulder (63.2 vs. 36.8; *p* < 0.001) and right shoulder (64.2 vs. 38.9; *p* < 0.001). This variable reported a large effect size for both shoulders, favoring males (Cohen’s d = 2.77 [left] and Cohen’s d = 2.47 [right]). There were no significant differences between males and females for time-to-peak force, and the effect size was small for both shoulders (Cohen’s d = 0.39 [left] and Cohen’s d = 0.01 [right]).

Table 2 contains the descriptive analysis of the ground reaction force peak and the range of the center of pressure displacement in the anteroposterior and lateral axes for each of the three tests performed. Likewise, the number of push-ups exerted during the second task is shown in this table, with significant differences between sexes (25.1 vs. 12.7; *p* < 0.001) in this variable, as well as in ground reaction force in all testing procedures in both arms (*p* < 0.001). The effect size was large, favoring males for the number of push-ups (Cohen’s d = 2.24) and the ground reaction force in the static push-up (Cohen’s d = 2.34 [left] and Cohen’s d = 2.49 [right]), counter-movement push-up (Cohen’s d = 3.21 [left] and Cohen’s d = 2.91 [right]), and jumping push-up (Cohen’s d = 1.63 [left] and Cohen’s d = 1.36 [right]). Nevertheless, regarding stability, there were no differences in the lateral and anteroposterior center of pressure displacement.

Table 3 describes the variables evaluated by the motion capture system, including mean and standard deviation in shoulder and elbow joint angles, as well as in humerus and forearm acceleration. As for shoulder elevation and elbow flexo-extension, the data obtained did not show differences between males and females (*p* > 0.05). However, males showed greater values in humerus acceleration than females both in the counter-movement push-up and jumping push-up tests (*p* < 0.05). The effect size was also large for these variables in the counter-movement push-up (Cohen’s d = 1.20 [left] and Cohen’s d = 1.10 [right]) and jumping push-up (Cohen’s d = 1.93 [left] and Cohen’s d = 1.57 [right]). Also, forearm acceleration was faster among males than females in the jumping push-up test (*p* < 0.001), with a large effect size (Cohen’s d = 2.55 [left] and Cohen’s d = 2.14 [right]), but not in the counter-movement push-up test (*p* > 0.05). Moreover, in terms of the percentage of decrement of acceleration and deceleration, no differences were found in the humerus or forearm in any test between sexes (*p* > 0.05).

A graphical summary of the main variables assessed by force plates and motion capture systems is presented in Figure 2 and Figure 3. The line represents the median value of the entire sample for each variable across time, while the shading indicates the envelopes. Concretely, lighter shading shows the full envelope of the data, ranging from maximum to minimum values at each time point, while darker shading indicates the interquartile range, from the 25th percentile to the 75th percentile, to highlight that the middle 50% of the data lies within.

Furthermore, Table 4 shows the analysis of the relationships found between the data obtained with the dynamometer and the variables extracted from the force plates. In regard to ground reaction force, this variable in all push-up variations was directly correlated with lateral abduction force peak in both arms (*p* < 0.05). However, this variable did not show significant relationships with the center of pressure displacement. Likewise, the time to peak was not correlated with any variable extracted from force plates. As for the lateral abduction force peak, this variable showed great direct relationships with the number of push-ups exerted during the counter-movement push-up test.

Moreover, Table 5 displays the regression analysis performed between the variables assessed by the force plates and the data obtained by the motion capture system. Regarding ground reaction force, this parameter was directly correlated with humerus and forearm acceleration both during the counter-movement push-up and the jumping push-up tests (*p* < 0.05), but not with shoulder elevation and elbow flexo-extension range of motion (*p* > 0.05). As for the center of pressure lateral displacement (COP_X) and anteroposterior displacement (COP_Y), these variables were directly correlated with the percentage of the decrement of humerus and forearm acceleration during the counter-movement push-up test (*p* < 0.05) and the deceleration of these segments during the eccentric phase of the jumping push-up test (*p* < 0.05).

Finally, Table 6 shows the regression models developed to estimate humerus acceleration using ground reaction force and center of pressure displacement data. These models were significant for the counter-movement push-up test in both arms (*p* < 0.001) and for the jumping push-up test in the left upper limb (*p* = 0.002) but not in the right upper limb (*p* = 0.086). These models were evaluated using cross-validated regression metrics, showing that the coefficient of determination (R^2^) indicated that approximately 80% of the variance in the dependent variable was explained by the predictors in the models concerning the counter-movement push-up tests, while this coefficient varied between 33% and 58% in the models concerning the jumping push-up tests. Furthermore, the root mean squared error (RMSE) was calculated, observing between 0.42 and 0.56 m/s^2^ of prediction error. Therefore, the mean squared error (MSE) varied between 0.18 and 0.31 m^2^/s^2^. Finally, the model’s effect size, calculated using Cohen’s f^2^, showed a large effect size (f^2^ = 4.46) for the models concerning the counter-movement push-up. Moreover, the first model concerning the jumping push-up using variables from the left shoulder also presented a large effect size (f^2^ = 1.40). However, the last model developed obtained a minor value of the model’s effect size, but it was large too (f^2^ = 0.49) according to established benchmarks.

## 4. Discussion

The present research aimed to compare the data extracted from force plates with the results assessed by a motion capture system and a dynamometer to demonstrate the validity of force plates in estimating shoulder stability in highly demanding sports movements. Previous studies have shown the validity of force plates by comparing them to the kinematic results based on the “gold standard” motion capture technology [29]. However, our study also included the use of a dynamometer to directly examine shoulder kinetic data (i.e., isometric joint moment characteristics). As far as the authors are aware, the current study is the first study that combines three different objective tools in developing a complete analysis of kinematics and kinetics parameters during three variations of push-up tasks focusing on shoulder stability. Therefore, this study could provide critical information on how we determine shoulder stability during loaded upper-limb movements.

The main finding of the current study was the great estimation power of the ground reaction force and center of pressure displacement measured with force plates to predict humerus acceleration during continuous push-ups and plyometric push-ups, a finding that is very relevant due to the key role that acceleration plays in glenohumeral instability, as it has already been noted by several authors [30,31].

Regarding lateral abduction analysis, time to peak was very similar between sexes, a finding that is very consistent with the current scientific literature, considering that time-related variables are not sex-dependent but are more influenced by the neural drive speed [18,32] and could be modified by different training modalities, highlighting resistance training [33]. Nevertheless, lateral abduction strength was clearly greater among males, thus coinciding with what was stated by Nuzzo et al. [34], confirming that force differences between sexes are more noticeable in the upper limb due to the greater cross-sectional muscle area and the larger type II fiber areas.

As for vertical force exerted during the push-up tests, the differences found were consistent with the previous findings, with higher values in males than in females in the three tasks performed. These differences were found in the number of push-ups exerted in thirty seconds, too, which is in agreement with some scientific proofs that hypothesize about the higher physiological challenge of push-ups for females than males, based on previous studies that found these variations in terms of velocity, weight shifted on hand contact points, and muscle activation [35,36]. However, instead of the higher capacity of males to perform push-ups, females are able to maintain good stability of the center of pressure during the tests, with very similar data compared to males, both in the lateral and anteroposterior axis. This finding could be explained by the theory that shoulder stability without structural damage depends on functional factors such as neuromuscular control and the ability to manage muscle fatigue such as several researchers have stated in the past [37,38]. Furthermore, alluding to this variable, the center of pressure displacement showed clear differences among the three push-up variations, with the minor values in the static test and the higher values in the jumping push-up test. This may be explained by the effects of gravity on the body mass of the subjects when they are in the airdrop phase, making them receive all their weight on their upper limbs, as has been previously stated by other authors [39,40].

In regard to the variables extracted from the motion capture system, joint angles seemed to be very similar among males and females and were almost constant among the two dynamic tests. Moreover, the range of motion observed both in shoulder elevation and elbow flexo-extension was consistent with previous scientific studies alluding to push-up analysis [41]. In addition, as for humerus and forearm acceleration, the results showed differences between sexes, thus agreeing with previous studies that established the major velocity and acceleration capabilities in sports tasks involving upper limbs among males [42]. Nonetheless, both sexes presented a similar percentage of decrement of acceleration during the thirty-second task, a finding that could be explained because the tests performed in this study are functional tests involving all the periarticular shoulder muscles, and instead of sex is considered a factor that influences peripheral fatigue in some specific muscles, these differences have not been observed in the general level of fatigue in previous studies [43].

As for the linear regression analysis between force assessed by the dynamometer and variables measured with force plates, a direct relation was found between lateral abduction torque and ground reaction force, thus coinciding with the previously established excellent intra-class correlation coefficient between dynamometers and force plates in shoulder force production assessment [44]. In addition, this dynamometric variable was directly correlated with the number of push-ups exerted during the counter-movement push-up test, a finding that matches the results exposed by Kotarsky et al. [45], who demonstrated that push-ups and strength training produce a mutual benefit between these two variables.

Finally, alluding to the linear regression analysis of the variables measured with the force plates and the motion capture system, the findings showed that the vertical force exerted during the dynamic push-up tests was directly related to the humerus and forearm acceleration. This finding is very relevant to the purpose of the present study due to these two parameters being clearly altered after a shoulder instability episode [46], so this relationship may be used to develop estimation equations using force as a predictor of instability or risk of instability. Furthermore, the center of pressure displacement in the *X*-axis and *Y*-axis showed a direct correlation with the percentage of humerus and forearm acceleration decrement, just again leading to the theory that force plates could be a reliable tool to indirectly estimate shoulder stability because the acceleration decrease is one of the main parameters used to estimate kinematic fatigue, which is a key variable in the analysis of shoulder instability injuries [8,47].

In summary, considering all the results found in the present study, it is confirmed that force plates are a valid tool in shoulder stability assessment in healthy active adults, and could be considered a promising tool to conduct evaluations among high-performance athletes that need to accomplish sports movements that generate a high demand in the neuromuscular control of the periarticular shoulder muscles in terms of ensure joint stability, such as other authors have demonstrated in lower limb instability-related injuries [48,49].

### Strengths and Limitations

The main limitation of this study is the wide heterogeneity among the participants regarding the push-up technique. Although this issue is not controllable, the same instructions were given to the participants before starting the testing procedure for hand positions, push-up depth, shoulder and elbow initial angles, and thorax and pelvis positions during the task to reduce this variability as much as possible. In addition, the physiological differences between sexes in this specific sport gesture may compromise the validity of the conclusions, so this variable was considered as a confounding variable during the statistical analysis. Moreover, the small number of participants enrolled could make it difficult to extract consistent results; however, the decision to include three variations of the push-up test may increase the consistency of the data extracted, allowing the selection of different evaluation procedures depending on the target population. Finally, the small sample size together with the imbalance in sex distribution also limited the consistency of the conclusions about sex comparisons, so the conclusions should be taken carefully and need to be confirmed in a larger population with a balanced distribution between sexes. Finally, the racial component could be a limitation too, since this study could not use this variable as a confounding variable due to the imbalance in ethnicities among the participants included. Further studies with a larger population should be developed to establish the differences in glenohumeral biomechanics based on the racial component.

## 5. Conclusions

The results of this research confirmed that healthy active adults present similar movement patterns during push-ups, without differences in terms of stability between sexes, although males showed greater values in lateral abduction strength and vertical force peak. Moreover, the current study demonstrated that force plates are a valid tool for upper-limb assessment with meaningful correlations with dynamometer and motion capture systems. Finally, this investigation allowed the development of stability estimation models, paving the way for a new model of shoulder evaluation in physically active adults. Further research is needed to demonstrate whether this tool could also be implemented to assess athletes who need to return to play highly demanding sports movements.

## Figures and Tables

**Figure 1 sensors-25-05333-f001:**
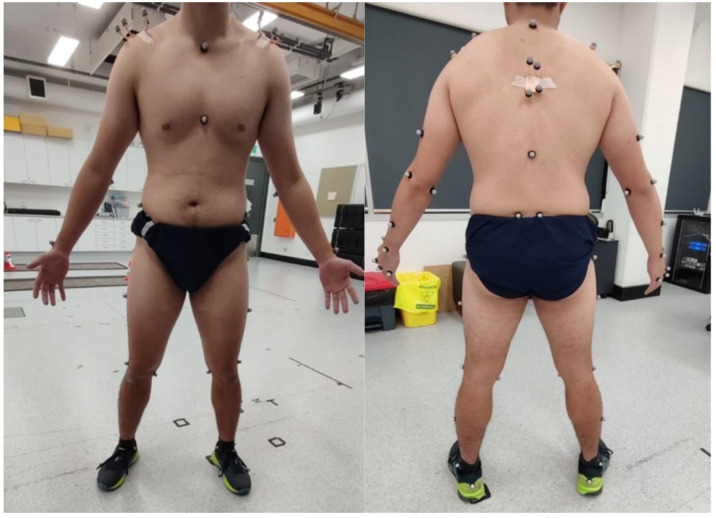
Reflective marker locations.

**Figure 2 sensors-25-05333-f002:**
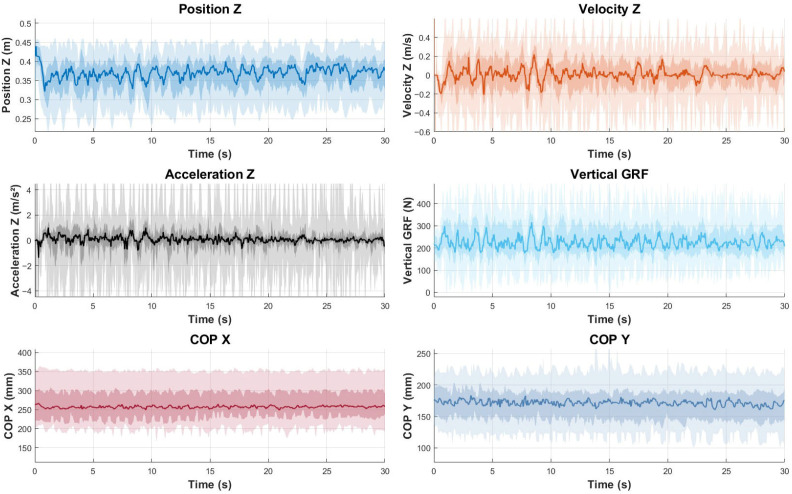
Graphical representation of the left upper-limb variables during the counter-movement push-up test for the entire sample (median values and envelopes). Units: Position *Z*-axis (m), Velocity *Z*-axis (m/s), Acceleration *Z*-axis (m/s^2^), Force *Z*-axis (Newtons), COP *X*-axis (mm), and COP *Y*-axis (mm). The horizontal axis for all of the graphs represents time in seconds.

**Figure 3 sensors-25-05333-f003:**
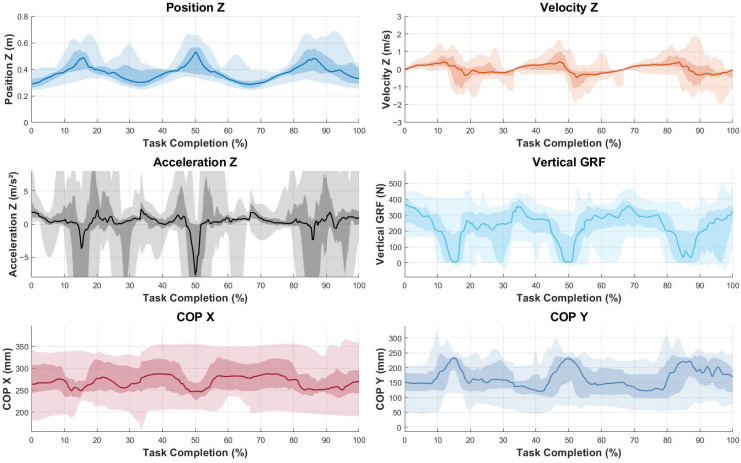
Graphical representation of the left upper-limb variables during the jumping push-up test for the entire sample (median values and envelopes). Units: Position *Z*-axis (m), Velocity *Z*-axis (m/s), Acceleration *Z*-axis (m/s^2^), Force *Z*-axis (Newtons), COP *X*-axis (mm), and COP *Y*-axis (mm). The horizontal axis for all of the graphs represents time in percentage (%).

**Table 1 sensors-25-05333-t001:** Descriptive analysis depends on the sex of the strength evaluated by the dynamometer.

Side	Force Peak		Confidence Interval (95%)		Mean Differences	Test Statistic	Effect Size (Cohen’s d)	Time to Peak		Confidence Interval (95%)		**Mean** **Differences**	**Test** **Statistic**	**Effect Size (Cohen’s d)**
	M	F	M	F				M	F	M	F			
LEFT	63.2 (9.42)	36.8 (9.67)	57.7–68.6	26.7–47.0	26.3 (4.63)	F = 31.6; *p* < 0.001	2.77	2.07 (1.05)	1.70 (0.79)	1.47–2.68	0.87–2.52	0.38 (0.48)	F = 0.79; *p* = 0.391	0.39
RIGHT	64.2 (10.75)	38.9 (9.75)	58.0–70.4	28.7–49.2	25.3 (5.11)	F = 26.6; *p* < 0.001	2.47	2.35 (1.07)	2.34 (1.24)	1.74–2.97	1.04–3.64	0.01 (0.55)	F = 0.0004; *p* = 0.984	0.01

M: male; F: female; and F and *p*: statistical proofs. UNITS: Force Peak is measured in Newtons; Time to Peak is measured in seconds.

**Table 2 sensors-25-05333-t002:** Descriptive analysis depends on the sex of the variables obtained by the force plates.

Task	Variable	Side	Outcomes		Confidence Interval (95%)		Test Statistic	Effect Size (Cohen’s d)
			M	F	M	F		
STATIC PUSH-UP	Ground Reaction Force	Left	272.53 (30.11)	205.65 (26.88)	255–290	177–234	F = 24.15; *p* < 0.001	2.34
		Right	288.03 (33.08)	217.02 (22.96)	269–307	193–241	F = 30.38; *p* < 0.001	2.49
	COP_X Displacement	Left	5.17 (0.83)	8.03 (2.78)	4.69–5.65	5.12–10.9	F = 6.11; *p* = 0.053	1.39
		Right	3.66 (0.77)	5.41 (0.31)	4.47–5.67	1.25–18.0	F = 52.26; *p* < 0.001	2.98
	COP_Y Displacement	Left	5.07 (1.04)	9.65 (8.00)	3.22–4.11	5.08–5.74	F = 1.95; *p* = 0.220	0.80
		Right	3.87 (0.46)	5.77 (1.93)	3.60–4.13	3.75–7.80	F = 5.74; *p* = 0.060	1.35
COUNTER-MOVEMENT PUSH-UP	Number of push-ups	Both	25.1 (6.88)	12.7 (3.72)	21.2–29.1	8.76–16.6	F = 27.34; *p* < 0.001	2.24
	Ground Reaction Force	Left	443.2 (70.19)	258.8 (40.86)	403–484	216–302	F = 54.02; *p* < 0.001	3.21
		Right	453.1 (70.04)	275.2 (50.80)	413–494	222–329	F = 40.56; *p* < 0.001	2.91
	COP_X Displacement	Left	21.0 (18.38)	17.8 (13.09)	10.4–31.6	4.08–31.6	F = 0.19; *p* = 0.668	0.20
		Right	12.6 (7.40)	11.9 (7.18)	11.7–29.7	4.85–36.3	F = 0.03; *p* = 0.859	0.10
	COP_Y Displacement	Left	20.7 (15.59)	20.6 (14.98)	8.28–16.8	4.38–19.4	F = 0.0003; *p* = 0.986	0.01
		Right	13.3 (8.12)	13.0 (8.00)	8.58–18.0	4.65–21.4	F = 0.003; *p* = 0.954	0.03
JUMPING PUSH-UP	Ground Reaction Force	Left	884.6 (323.83)	468.2 (151.59)	698–1072	309–627	F = 15.32; *p* = 0.001	1.63
		Right	768.0 (266.51)	479.9 (136.19)	614–922	337–623	F = 10.17; *p* = 0.005	1.36
	COP_X Displacement	Left	35.0 (11.23)	30.1 (8.86)	28.5–41.5	20.8–39.4	F = 1.09; *p* = 0.316	0.48
		Right	30.0 (7.18)	25.0 (7.27)	43.1–75.3	19.5–62.4	F = 2.02; *p* = 0.187	0.69
	COP_Y Displacement	Left	59.2 (27.84)	40.9 (20.44)	25.9–34.1	17.3–32.6	F = 2.67; *p* = 0.126	0.75
		Right	33.3 (15.25)	25.6 (5.31)	24.5–42.1	20.0–31.1	F = 2.80; *p* = 0.112	0.67

COP_X: Centre of Pressure in *X*-axis (lateral displacement); COP_Y: Centre of Pressure in *Y*-axis (anteroposterior displacement M: Male; F: Female; and F and *p*: statistical proofs. UNITS: Ground Reaction Force is measured in Newtons; COP_X and COP_Y Displacement is measured in millimeters.

**Table 3 sensors-25-05333-t003:** Descriptive analysis depends on the sex of joint angles and upper-limb acceleration assessed by the motion capture system.

Task	Variable	Side	Outcomes		Confidence Interval (95%)		Test Statistic	Effect Size (Cohen’s d)
			M	F	M	F		
COUNTER-MOVEMENT PUSH-UP	Shoulder Elevation ROM	Left	43.32 (11.2)	46.04 (13.03)	36.9–49.8	32.4–59.7	F = 0.197; *p* = 0.668	0.22
		Right	45.33 (11.16)	47.33 (13.54)	38.9–51.8	33.1–61.5	F = 0.102; *p* = 0.758	0.16
	Elbow Flexo-Extension ROM	Left	122.34 (41.58)	104.77 (54.86)	98.3–146	47.2–162	F = 0.494; *p* = 0.503	0.36
		Right	126.14 (45.01)	108.30 (49.04)	100–152	56.8–160	F = 0.583; *p* = 0.465	0.38
	Humerus Max. Acc.	Left	2.22 (0.94)	1.22 (0.70)	1.68–2.77	0.49–1.96	F = 6.932; *p* = 0.021	1.20
		Right	2.21 (1.00)	1.20 (0.81)	1.63–2.79	0.36–2.05	F = 5.603; *p* = 0.036	1.10
	Humerus Acc. Decrement (%)	Left	46.63 (16.47)	43.26 (20.26)	37.1–56.1	22.0–64.5	F = 0.130; *p* = 0.728	0.18
		Right	44.90 (16.36)	44.45 (15.73)	35.4–54.3	27.9–61.0	F = 0.003; *p* = 0.955	0.03
	Forearm Max. Acc.	Left	0.67 (0.24)	0.49 (0.33)	0.53–0.81	0.15–0.83	F = 1.456; *p* = 0.264	0.62
		Right	0.62 (0.19)	0.89 (1.06)	0.51–0.72	−0.21–2.00	F = 0.409; *p* = 0.550	0.35
	Forearm Acc. Decrement (%)	Left	59.64 (16.69)	64.10 (13.13)	50.0–69.3	50.3–77.9	F = 0.409; *p* = 0.534	0.29
		Right	58.53 (17.13)	71.00 (13.52)	48.6–68.4	56.8–85.2	F = 3.027; *p* = 0.107	0.81
JUMPING PUSH-UP	Shoulder Elevation ROM	Left	57.00 (17.02)	53.54 (27.01)	47.2–66.8	25.2–81.9	F = 0.084; *p* = 0.780	0.15
		Right	53.54 (16.80)	50.65 (20.27)	43.6–63.0	29.3–72.0	F = 0.077; *p* = 0.788	0.16
	Elbow Flexo-Extension ROM	Left	132.33 (31.78)	99.35 (51.23)	114–151	45.6–153	F = 2.135; *p* = 0.189	0.77
		Right	125.72 (40.69)	103.23 (50.16)	102–149	50.6–156	F = 0.942; *p* = 0.360	0.49
	Humerus Max. Acc.	Left	1.90 (0.64)	0.90 (0.35)	1.53–2.27	0.53–1.26	F = 20.488; *p* < 0.001	1.93
		Right	1.91 (0.62)	1.04 (0.48)	1.56–2.27	0.54–1.54	F = 11.900; *p* = 0.005	1.57
	Humerus Max. Dec.	Left	0.16 (0.51)	−0.16 (0.38)	−0.13–0.46	−0.56–0.23	F = 2.509; *p* = 0.138	0.71
		Right	0.28 (0.53)	−0.07 (0.39)	−0.02–0.59	−0.48–0.34	F = 2.690; *p* = 0.125	0.75
	Forearm Max. Acc.	Left	2.85 (0.91)	1.06 (0.40)	2.33–3.38	0.64–1.48	F = 37.160; *p* < 0.001	2.55
		Right	2.70 (0.75)	1.22 (0.63)	2.26–3.13	0.56–1.89	F = 20.342; *p* < 0.001	2.14
	Forearm Max. Dec.	Left	−0.14 (0.17)	−0.04 (0.08)	−0.24–−0.04	−0.12–0.04	F = 3.028; *p* = 0.099	0.75
		Right	−0.13 (0.10)	−0.07 (0.11)	−0.19–−0.07	−0.18–0.05	F = 1.501; *p* = 0.252	0.57

UNITS: Shoulder Elevation and Elbow Flexo-Extension ROM (range of motion) are measured in degrees; Humerus and Forearm Maximum Accelerations and Decelerations are measured in m/s^2^, and Humerus and Forearm Acceleration Decrements are measured as a percentage of the lack of acceleration considering the maximum acceleration as reference.

**Table 4 sensors-25-05333-t004:** Analysis of the correlations between the variables assessed by the dynamometer and force plates.

Task	Variable	Side	Force Peak (Left)	Force Peak (Right)	Time to Peak (Left)	Time to Peak (Right)	GRF (Left)	GRF (Right)
STATIC PUSH-UP	Ground Reaction Force	Left	0.667 **	0.731 ***	−0.164	−0.273	-	0.799 ***
		Right	0.726 ***	0.750 ***	0.006	−0.216	0.799 ***	-
	COP_X Displacement	Left	−0.512 *	−0.551 *	0.223	0.226	−0.804 ***	−0.442
		Right	−0.827 ***	−0.841 ***	0.009	0.167	−0.839 ***	−0.878 ***
	COP_Y Displacement	Left	−0.270	−0.326	0.220	0.115	−0.614 **	−0.194
		Right	−0.443	−0.472 *	0.083	0.110	−0.725 ***	−0.396
COUNTER-MOVEMENT PUSH-UP	Number of push-ups	Both	0.598 **	0.553 *	0.379	0.276	0.730 ***	0.713 ***
	Ground Reaction Force	Left	0.759 ***	0.747 ***	0.086	0.173	-	0.993 ***
		Right	0.749 ***	0.761 ***	0.065	0.140	0.993 ***	-
	COP_X Displacement	Left	0.020	0.003	−0.121	0.222	0.353	0.355
		Right	−0.010	−0.006	−0.064	0.251	0.430	0.444
	COP_Y Displacement	Left	−0.079	−0.089	−0.207	0.191	0.366	0.382
		Right	−0.015	−0.043	−0.124	0.208	0.406	0.414
JUMPING PUSH-UP	Ground Reaction Force	Left	0.714 ***	0.641 **	−0.034	−0.119	-	0.832 ***
		Right	0.615 **	0.519 *	−0.033	−0.116	0.832 ***	-
	COP_X Displacement	Left	0.265	0.159	0.216	0.196	−0.090	−0.074
		Right	0.439	0.358	0.128	0.070	0.458 *	0.394
	COP_Y Displacement	Left	0.538 *	0.467 *	0.167	0.200	0.167	0.120
		Right	0.172	0.316	−0.185	−0.229	0.164	0.208

COP_X: Centre of Pressure in *X*-axis (lateral); COP_Y: Centre of Pressure in *Y*-axis (anteroposterior); and GRF: Ground Reaction Force (in each task). * *p* < 0.05; ** *p* < 0.01; and *** *p* < 0.001.

**Table 5 sensors-25-05333-t005:** Analysis of the correlations between the variables assessed by the motion capture system and force plates.

Task	Variable	Side	GRF (Left)	GRF (Right)	COP_X (Left)	COP_X (Right)	COP_Y (Left)	COP_Y (Right)
COUNTER-MOVEMENT PUSH-UP	Shoulder Elevation ROM	Left	−0.170	−0.194	0.203	0.172	0.318	0.260
		Right	−0.153	−0.187	−0.386	−0.337	−0.249	−0.241
	Elbow Flexo-Extension ROM	Left	0.302	0.344	0.021	0.149	0.087	0.053
		Right	0.285	0.312	−0.010	0.129	0.053	0.023
	Humerus Max. Acc.	Left	0.679 **	0.677 **	0.781 ***	0.739 ***	0.678 **	0.684 ***
		Right	−0.084	−0.109	−0.225	−0.138	−0.017	−0.089
	Humerus Acc. Decrement (%)	Left	0.721 ***	0.726 ***	0.761 ***	0.801 ***	0.729 ***	0.743 ***
		Right	−0.115	−0.155	−0.255	−0.169	−0.149	−0.095
	Forearm Max. Acc.	Left	0.477 *	0.474 *	0.578 **	0.539 *	0.528 *	0.508 *
		Right	−0.063	−0.059	0.414	0.458 *	0.388	0.389
	Forearm Acc. Decrement (%)	Left	−0.007	0.010	0.205	0.329	0.394	0.433
		Right	−0.313	−0.318	0.075	0.143	0.205	0.273
JUMPING PUSH-UP	Shoulder Elevation ROM	Left	0.052	0.174	0.031	0.308	−0.089	0.122
		Right	0.069	0.238	−0.057	0.374	0.021	0.243
	Elbow Flexo-Extension ROM	Left	0.394	0.350	0.339	0.075	0.464 *	0.098
		Right	0.276	0.268	0.468 *	0.104	0.506 *	0.088
	Humerus Max. Acc.	Left	0.626 **	0.640 **	0.373	0.366	0.396	−0.056
		Right	0.490 *	0.545 *	0.431	0.318	0.436	0.051
	Humerus Max. Dec.	Left	−0.235	−0.012	0.450 *	−0.140	0.272	−0.250
		Right	−0.198	0.072	0.539 *	−0.104	0.331	−0.193
	Forearm Max. Acc.	Left	0.628 **	0.628 **	0.460 *	0.405	0.463 *	0.019
		Right	0.50 *	0.556 *	0.501 *	0.348	0.529 *	0.132
	Forearm Max. Dec.	Left	0.015	0.003	−0.462 *	−0.193	−0.573 **	−0.003
		Right	−0.078	−0.069	−0.597 **	−0.262	−0.701 ***	0.139

COP_X: Centre of Pressure in *X*-axis (lateral); COP_Y: Centre of Pressure in *Y*-axis (anteroposterior); and GRF: Ground Reaction Force (in each task). * *p* < 0.05; ** *p* < 0.01; and *** *p* < 0.001.

**Table 6 sensors-25-05333-t006:** Prediction models to estimate motion capture stability from the force plate data.

Model	Dependent Variable	Independent Variables	Estimator	R	R^2^	AIC	BIC	RMSE	F	*p*
#1	Maximum Humerus Acceleration—Left [CMJ]	Constant	−0.41601	0.896	0.802	32.3	37.3	0.423	21.6	<0.001
		GRF Left	0.00432 ***							
		COP_X Left	0.04522 **							
		COP_Y Left	−0.01166							
#2	Maximum Humerus Acceleration—Right [CMJ]	Constant	−0.97582	0.904	0.817	33.4	38.4	0.434	23.8	<0.001
		GRF Right	0.00454 ***							
		COP_X Right	0.11994 *							
		COP_Y Right	−0.03120							
#3	Maximum Humerus Acceleration—Left [JUMPING]	Constant	−0.57262	0.763	0.583	35.7	40.7	0.460	7.45	0.002
		GRF Left	0.00149 ***							
		COP_X Left	0.03652							
		COP_Y Left	−0.00338							
#4	Maximum Humerus Acceleration—Right [JUMPING]	Constant	0.46111	0.575	0.330	43.5	48.5	0.559	2.63	0.086
		GRF Right	0.00130							
		COP_X Right	0.02044							
		COP_Y Right	−0.00892							

COP_X: Centre of Pressure in *X*-axis (lateral); COP_Y: Centre of Pressure in *Y*-axis (anteroposterior); CMJ: counter-movement push-up; and GRF: Ground Reaction Force (in each task). * *p* < 0.05; ** *p* < 0.01; and *** *p* < 0.001.

## Data Availability

The data are available upon reasonable request to the corresponding author.

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
