# Peer review of "In Vivo (In)Stability Shoulder Assessment in Healthy Active Adults Using Force Plates and a Motion Capture System: A Cross-Sectional Study"

_sensors, 2025, doi:10.3390/s25175333_

Round 1
Reviewer 1 Report
Comments and Suggestions for Authors
This study recruited 20 healthy, physically active adults to examine whether data from bilateral force plates collected during three push‑up variations (static, countermovement, and plyometric) can serve as a proxy for glenohumeral (in)stability. The authors conclude that force plates offer a valid, objective tool to estimate shoulder stability in the tested population.
- Its content does not fit in the special issue’s scope, which is seeking contributions that focus on new sensor designs, novel uses for existing sensors, optimizing outputs or signal processing techniques to improve the usefulness of sensors, and evaluating sensors in real-world settings (https://www.mdpi.com/journal/sensors/special_issues/Z99627YINZ). MDPI Sensors typically expects a technological or methodological contribution beyond applying well‑established devices (in this case, commercial force plates and 3‑D motion capture). The study would better fit journals specialising in sports biomechanics or orthopaedic rehabilitation, where the primary interest is physiological interpretation rather than sensor innovation.
- Strengthen the technological contribution if resubmitting to a sensor‑focused journal. For example:
-
- Develop and validate a novel algorithm for real‑time COP‑instability detection on low‑cost force platforms.
- Introduce sensor‑fusion (e.g., IMU + force‑plate) pipelines that outperform single‑sensor approaches.
- Provide complete sensor specifications (model, sampling rate, calibration), signal‑processing steps (filter types, cut‑off frequencies), and push‑up standardisation protocols.
- include effect sizes, confidence intervals, correction for multiple testing, and cross‑validated regression metrics.
- Adopt the conventional IMRaD structure, move key figures/tables into the main text, and streamline the narrative.
Reviewer 2 Report
Comments and Suggestions for Authors
The study presented considers the evaluation through a new model using force platform to evaluate the stability of shoulder from push-up exercises in high performance athletes. Although the article presents the results and the methodologís properly, it must improve in some important aspects:
Jamovi 2.4.11 from what company, university or development is the platform.
Paragraph from line 183 until 189 should be in a vertical page. In general horizontal pages are exclusive of tables or very big images.
The figures 2 and 3 do not have the units of measure in the vertical or horizontal axis. It's ok that is in the legend, but I don’t know if the time is in seconds or not.
It concludes that this could be an useful tool for the evlaución of shoulder stability in high performance athletes, but in the sample it does not incorporate these subjects. Why from the results obtained comparing only technology conclude this?
The comparison between sexs can not be conclusive. This is because the little quantity of subjects measured in which it does not consider the covariable of sex, duplicating its statistically significant sample.
Improving these aspects and clarifing the limitations and conclusioes of the study it could be in conditions of be published from my point of view.
Reviewer 3 Report
Comments and Suggestions for Authors
The overall impression of the paper is good – the authors provide a good overview of modern research in this area, including elements of critical analysis and comparison with their work. The results of the study confirm the conclusions and are consistent with the abstract. However, there are a number of issues that need to be and are useful to address before publication:
- It is not clear from the abstract what the direction of the work concept is – rehabilitation (if so, what type of sports), involving healthy adults in sports etc.
- A strange exclusion criterion for not knowing English. If knowing English directly affects the motor functions of the shoulder joint, then the authors should provide such a link. Otherwise, it is worth writing that the selection was initially conducted among volunteers who knew English.
- For section 2, it is preferable to accompany the text with a flow chart for excluding subjects from the sample.
- ICC – must be deciphered before and at the moment of first use in the text.
- For each measuring device described in paragraphs 2.2.2, 2.2.3, it is necessary to present the error of the measurements performed.
- Despite the fact that the authors reasonably indicate the method of placing marks on the body, as well as additional marks, it is strange to me that such large muscles as the pectoralis major are not taken into account when studying the biomechanics of the shoulder joint. In my opinion, the authors should comment on why sensors are not installed there.
- Line 142 - the authors write about calculating Pearson correlations, then it is necessary to indicate which test(s) were used to check the normality of the measured values. If any of the values is not normally distributed, other methods of analysis should be used.
- The literature contains many examples of racial differences in the biomechanics of joints, vascular walls, rheumatic diseases, etc. The authors should pay closer attention to the racial component of the differences obtained in the study. While I cannot assess the impact of race on the outcome in this study (there are several isolated racial inclusions), the authors should conduct a literature review on racial differences in shoulder biomechanics (or lack of publications in this area).
- Tab 1,2,3 – units of measurement are needed.
- For Fig. 2, the legend - increase the font or move it to the caption to the picture. Description for Fig. 2, add that it is for a specific volunteer (and his identifier in brackets).
- For Fig. 3, the remark is the same as for Fig. 2. In general, I would like to see a different presentation of the data for the entire group. A fairly standard presentation of the data is in the form of envelopes (min, max) of values and the median for the entire sample.
Round 2
Reviewer 2 Report
Comments and Suggestions for Authors
The authors addressed all the comments. The form of the article was notably improved.